

# Predicting oceanic Lagrangian trajectories with hybrid space-time CNN architecture

Lorenzo Della Cioppa[1] and Bruno Buongiorno Nardelli[1]

[1]Consiglio Nazionale delle Ricerche, Istituto di Scienze Marine (CNR-ISMAR), 80133 Naples, Italy

**Correspondence:** Lorenzo Della Cioppa (lorenzo.dellacioppa@cnr.it)

**Abstract.** Lagrangian dynamics simulation is a challenging task, as it typically depends on integrating velocity fields, whose estimation is inherently difficult due to both theoretical and technical constraints. Neural Network approaches provide practical methods to overcome most of related complications by learning directly from data. In this paper a deep Convolutional Neural Network (CNN) for Lagrangian trajectories simulation is presented. The proposed architecture is inspired by existing Computer Vision methods, combining Long-Short Term Memory and U-Net architectures to enforce causality. Several training setups are considered, including conditional Generative Adversarial Network (cGAN) training. The results are evaluated using Lagrangian metrics.

## 1 Introduction

The analysis of Lagrangian trajectories provides a fundamental contribution to the study of ocean dynamics. Even though satellite observations and numerical general circulation models have been significantly progressing in the estimation of global ocean surface currents, in situ Lagrangian drifters still represent the most important source of ground-truth measurements ubiquitously available. Lagrangian observations provide a complementary view with respect to that obtained from satellite altimeter data. In fact, due to both technical and theoretical limitations, altimeter-based currents are unable to fully describe mesoscale and submesoscale dynamics Molcard et al. (2003); Aksamit et al. (2020); Lacorata et al. (2001, 2014); Corrado et al. (2017). Velocities derived from in situ drifters, instead, do include all dynamical components acting at the ocean surface. Simulation of Lagrangian trajectories is also of great importance, being instrumental in several practical and scientific applications, such as the tracking and prediction of pollutants and debris positions at sea, or the transport and dispersal of small marine organisms. Indeed, a strong interest has recently grown around the concept of sea connectivity, which is fundamentally defined in a Lagrangian framework Ser-Giacomi et al. (2021), opening new perspectives on the theoretical foundations of marine ecosystem functioning. In addition, Lagrangian trajectories are necessary in the study of dispersion properties Corrado et al. (2017) and they provide ground truth for the validation of numerical circulation models Lacorata et al. (2014).





Conversely to the Eulerian approach, where quantities of interest are defined/measured on a fixed spatial grid, Lagrangian techniques directly follow the evolution of relevant variables along the trajectories resulting from the advection of passive buoys (drifters) , whether real or simulated ones. Despite the large number of deployed in situ drifters, their number is generally not sufficient to provide a regular space/time sampling and they are not sufficient for any practical operational trajectory prediction task. Moreover, drifter deployment can be expensive and time consuming, and cannot cover all zones of interest (e.g. remote ocean regions). As a result, numerical experiments using simulated trajectories are widely used for several applications requiring Lagrangian analyses.

The most used technique to produce synthetic Lagrangian trajectories is to integrate the appropriate velocity fields. Velocity fields estimation is not without difficulties itself. Sea surface velocities derived from satellite altimeters data assume balanced geostrophic motion. However, the geostrophic approximation is only accurate to first order and cannot provide accurate velocities when applied to evolving features at the meso/submesoscale. Moreover it is not possible to apply geostrophic balance directly in the equatorial zones, where the Coriolis force is zero. Geostrophic velocities thus leave out fundamental components that drive the evolution of the system. In addition, satellite altimeter coverage is still low both in time and space, and ad-hoc methods must be implemented to be able to monitor the evolution of the velocity fields on a regular latitude-longitude grid. On the other hand, synthetic velocity fields produced by high-resolution simulations depend significantly on the specific model at hand. Nowadays, many simulations assimilate real data and/or are forced by real data. It should be noted, however, that the accuracy of model surface velocities, evaluated through metrics that compare real drifter velocities and simulated ones, is generally lower than that obtained from satellite altimetry. Moreover, no model is currently capable of simulating Lagrangian trajectories and Lagrangian dispersion properties without error Lacorata et al. (2014). In the end, when task at hand requires Lagrangian simulations, the model is chosen as the one more suited for the required analyses (e.g. dispersion properties rather than long-time forecasting, etc.).

Only few methods have been proposed to assimilate the information of real drifters and directly correct available 2D velocity fields Molcard et al. (2003). More recently, machine learning approaches have been proposed to tackle different aspects of the Lagrangian simulation problem, i.e. either as alternatives to numerical advection schemes or to serve as a basis to learn true dynamical properties from observed drifters and to improve trajectory predictions. As an example, leveraging the Lagrangian-Eulerian duality, a U-net based method has been proposed to evolve probability distributions time step by time step, and then recovering the trajectory as the maximum of the probability at each time Jenkins et al. (2023). Due to time-space nature of the problem, convolutional long-short term memory (LSTM) neural network have been proposed in order to produce the whole trajectory, combining both initial position and velocity fields as the input of the network Botvynko et al. (2023). By using an inverse problem formulation, Botvynko et al. (2023) also suggested a new strategy to inform sea surface velocities from in situ drifter trajectories

In this paper we build upon the existing artificial intelligence approaches and propose a novel neural network architecture for Lagrangian trajectories simulation. Although the problem at hand obeys physical laws, we do not consider physically informed strategies as presently viable. The enforcing of physical properties should in fact be carried out with the knowledge of the velocity fields, which we want to avoid. Instead, we propose a hybrid U-net/LSTM design to enforce causality and preserve



the physical meaning of the output. A generator/discriminator approach is taken to further improve the results. Conversely to existing assimilation methods, which have to be ad-hoc designed, the proposed model allows for any observable Eulerian tracer to contribute to the trajectories generation.

The model is trained on a dataset of synthetic trajectories produced using sea surface height-derived currents in the Mediterrenean Sea, and validated on independent simulations from the same dataset. In addition to zonal and meridional velocities, sea surface temperature (sst) is used as a input of the proposed model to further improve performances.

## 1.1 Problem Statement

Given a velocity field $u(x,t) : \mathbb{R}^2 \times [t_0, +\infty) \to \mathbb{R}^2$, the evolution of a single particle passively advected by $u$ is described by the following Cauchy problem:

$$
\begin{cases}
\dfrac{d}{dt}x(t) = u(x(t),t) \\
x(t_0) = x_0
\end{cases}
. \tag{1}
$$

Tracers advected by $u$ can be described in terms of Lagrangian trajectories: the evolution of a tracer $\theta(t,x)$ from a time $t_0$ and a location $x_0$ to a time $t_1$, is given by $\theta(t_1, x(t_1))$, where $x(t)$ is a solution of 1.

That is, rather than measuring a quantity in fixed place and time, in the Lagrangian framework individual particles in the flow are followed, making possible to track the amount tracer each of them transports.

Although equivalent from a theoretical point of view, in practice Eulerian and Lagrangian frameworks are complementary in the study of particles and tracer dynamics: analysis of chaotic motion and dispersion regimes is only possible from a Lagrangian point of view, and the concept of sea connectivity is grounded in the Lagrangian view. Tracking and forecasting of pollutants is both more straightforward and computationally inexpensive in a Lagrangian framework. Moreover, simulating tracers evolution using Eulerian models is computationally demanding, whereas simulating Lagrangian trajectories is cheaper and faster.

The differential problem (1) is typically fully non-linear and the velocity field $u$ is not given in analytical form: whether the velocity fields are derived from geostrophic balance or produced by numerical hydrodynamic models, they are defined on a discrete grid with finite and often coarse resolution. Non-trivial algorithms are necessary to correctly interpolate the velocity fields on the curvilinear lat-lon grids typically used in oceanographic sciences van Sebille et al. (2018); Kehl et al. (2023); Delandmeter and van Sebille (2019). State of the art software for problem (1) in oceanographic context implement explicit integration schemes, often with integration step adaptivity Delandmeter and van Sebille (2019); Lange and van Sebille (2017); Kehl et al. (2023), typical choice being the Runge-Kutta 4th order scheme RK4 or the adaptive RK45 method Quarteroni et al. (2007).

## 1.2 Network description

Artificial Neural Networks (ANN) are a particular type of machine learning techniques that have proved capable of easily integrating available real world data into several different tasks, accompanying or replacing previous methods. The ANN is



trained for the specific task at hand by adjusting its internal parameters over the desired dataset. Various techniques exist to perform the training, and to help the network to generalize correctly.

Convolutional neural networks (CNN) are a powerful ANN architecture originally proposed for image processing applica-
90 tions, such as classification Krizhevsky et al. (2012), Bottou et al. (1994); Lecun et al. (1998), segmentation Ronneberger et al. (2015), image generation Goodfellow et al. (2020); Ho et al. (2020), etc. Whereas classical ANN's correlate each input datum with each other to produce the output, CNN's perform convolutions on the input image, meaning that, for any given pixel, only the neighboring pixels inside a given range are used in the computation. Due to their convolutional structure, CNN are cheaper than classical ANN in computer vision tasks and have several desirable properties, for example, they are naturally
shift-invariant.

As the components of Eulerian velocity fields defined on a regular structured grid can be considered images for any computational purpose, the use of CNN comes natural. In order to capitalize on the advantages of CNN architecture, the positions of the Lagrangian trajectories need to be suitably embedded into images.

### 1.2.1 The proposed simulation network

For the task of Lagrangian simulation, we propose a fully convolutional neural network architecture inspired by the well known pix2pix network Isola et al. (2017), a conditional generative adversarial neural network (cGAN) designed for image-to-image translation, including video frame prediction.

In the cGAN framework two different networks are used: a generator $G$, which produces the images, and a discriminator $D$, whose task is to decide whether an image is real or produced by the generator Goodfellow et al. (2020).
Faithful to the image-to-image translation paradigm of pix2pix, the proposed network aims at transforming a constant-in-time embedding of the initial position of the Lagrangian trajectory into the embedding of the fully evolved trajectory.

To this scope, we define two coupled operators of embedding/de-embedding $\mathcal{E} : \mathbb{R}^2 \to \mathbb{R}^{N \times N}$ and $\mathcal{E}^{-1} : \mathbb{R}^{N \times N} \to \mathbb{R}^2$, whose roles are, respectively, to translate the positions $x_t$ of a Lagrangian trajectory into a sequence of images, and to recover from those images the one-dimensional trajectory $x_t$.
Hence, assuming $x \in \mathbb{R}^2$ is a position, the output of $\mathcal{E}(x)$ is an image embedding the position $x$ and $\mathcal{E}^{-1}$ is such that $\mathcal{E}^{-1}(\mathcal{E}(x)) = x$. We choose to fix the embedding given by $\mathcal{E}$ in the form of a gaussian function, and to use a simple deep CNN as the de-embedder $\mathcal{E}^{-1}$, which is trained in advance to ensure the process is not biased by the trajectories positions. A more precise description of the embedding/de-embedding process is given in subsection 1.2.3.

The generator network $G$ takes as input a 5D tensor and outputs a 5D tensor, whose dimensions are **BTXYC** (**B**atch, **T**ime,
**X** dimension, **Y** dimension, **C**hannel). The input tensor is, for each time, the concatenation along the channels of the initial datum embedding, the meridional velocity field, the zonal velocity field and the temperature field. The output tensor contains a single channel and, for each time $t$, the corresponding 2D image is the embedding of the coordinates of the predicted trajectory at time $t$.

The architecture we use is a CNN based on the popular U-Net, which has been originally proposed for biomedical image
segmentation Ronneberger et al. (2015), but it has been since used for several different tasks, de-facto substituting autoencoders




as the go-to generator architecture in GANs and diffusion models. Similarly to autoencoders, U-Nets are composed of a contracting half (the encoder) and an expanding half (the decoder) where the role of the first is to compress the input into a smaller latent space and the role of the latter is to expand it back to the proper dimensions. Each level of the encoder downsamples the image by a factor of 2, while the corresponding decoder level upsamples it by a factor of 2. The main

difference with autoencoders is that U-Net architectures present skip connections between the contracting and the expanding halves of the network, which directly feed the outputs of each encoder layer into the corresponding decoder layer.

Our network deviates from the usual implementation of a U-net in three aspects: the downsample and upsample blocks are defined in a slightly different way (we don't use max pooling); due to the three-dimensional nature of the problem (one time dimension and two spatial dimensions), 3D convolutions are used instead of 2D ones; and the skip connections, rather than be

directly fed to the decoder, are processed by time-processing blocks. The latter aspect is, to the best knowledge of the authors, never been used before. The proposed architecture, along with the embedding/de-embedding system, is depicted in fig. 2. The upsample and downsample blocks are depicted in fig. 1.

Following Isola et al. (2017), batch normalization layers are added in the encoder block and dropout layers with probability $p = 1/2$ are added in the decoder. Based on empirical evaluations, swish activation functions are chosen instead of the original

leaky ReLu/ReLu. A last 3D convolutional block handles the aggregation of the different channels into the output image.

Although each channel of input data is three dimensional, only the spatial dimensions are downsampled/upsampled. The size of the kernels is 3 in space and 2 in time, as the long-time correlation are handled by the 2D convolutional LSTM, whose structure helps preserve causality across times.

The main rationale behind this design is to separate spatial processing from time processing: spatial processing (3D convolu-

140 tions) correlates the velocity fields with the trajectory position at each time and temporal processing (2D convolutional LSTM) correlates the current time state with the next one. Separation allows for each part to better specialize at its task.

The number of filters of the convolutions in each of the 3 levels is respectively 48, 64, 96. The total number of trainable parameters is 3134705.

The discriminator network is composed of the encoder part of the U-Net, with an additional sigmoid activation function to

145 normalize the output between 0 and 1. Conversely to the generator, for the discriminator the time dimension is donwsampled as well. The total number of trainable parameters is 452129.

### 1.2.2 Training setup

The proposed network has been trained using both GAN and non-GAN training setups. The total training setup is depicted in fig. 3.

Regarding the GAN training setup, the loss function design follows Isola et al. (2017). Let $\mathrm{H}(p, q)$ be the binary cross-entropy of $q$ from $p$, let $G$ be the generator, $D$ be the discriminator, $x$ sampled from input data and $y$ sampled accordingly from the target data. The cGan part of the loss can be expressed as





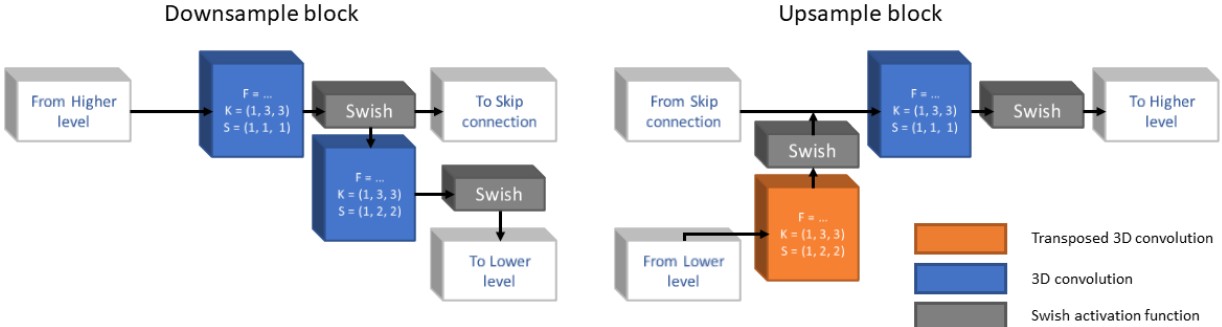

**Figure 1.** Downsample and upsample convolutional blocks for the proposed architecture. Kernel sizes and strides are indicated for each convolution, the number of filters depends on the level.

$$\mathcal{L}_{cGAN} = -\mathrm{H}(1, D(x,y)) - \mathrm{H}(0, D(x, G(x))) \tag{2}$$

$$= \mathbb{E}_{x,y}\left[\log(D(x,y)\right] + \mathbb{E}_x\left[\log(1 - D(x, G(x))\right]. \tag{3}$$

The cGAN loss goal is to train the discriminator to correctly classify both the output tensor of $G$ and the real target data. An $\ell^1$ term is added, multiplied by a constant $\lambda$, to handle low-frequency content. In addition, a time-dependent weight term $w(n) = n^{-s}$, $s = 0.5$ is used to increase accuracy in the first inference times. Hence the loss is

$$\mathcal{L} = \mathcal{L}_{cGAN} + \lambda \mathbb{E}_{x,y}\left[\|(G(x) - y) \cdot w\|_1\right]. \tag{4}$$

In our experiments, we set $\lambda = 0.2$.

The training procedure alternates an optimization step for the generator and one for the discriminator. The actual losses simplifies as we consider only the part of the loss that involves the parameters of the considered network (the gradient of the other being zero).

In the non-Gan case the training setup is the same as the GAN case, except that the cGAN loss is not used, and only the $\ell^1$ term is present.

The chosen optimizer is the AdamW optimizer Loshchilov and Hutter (2019), with learning rate $\eta = 2 \cdot 10^{-5}$, $\beta_1 = 0.9$, $\beta_2 = 0.999$, weight decay $\mu = 0.1$ and the AMSgrad option.

### 1.2.3 Embedder/de-embedder

In order to correctly handle the embedding, we decided to operate on the same grid of the velocity fields. We chose patches of size $24 \times 24$ and normalized the coordinates of the Lagrangian trajectories in the range $[0, 1)$ accordingly. Each normalized







**Figure 2.** Proposed generator architecture and embedding/de-embedding setup.





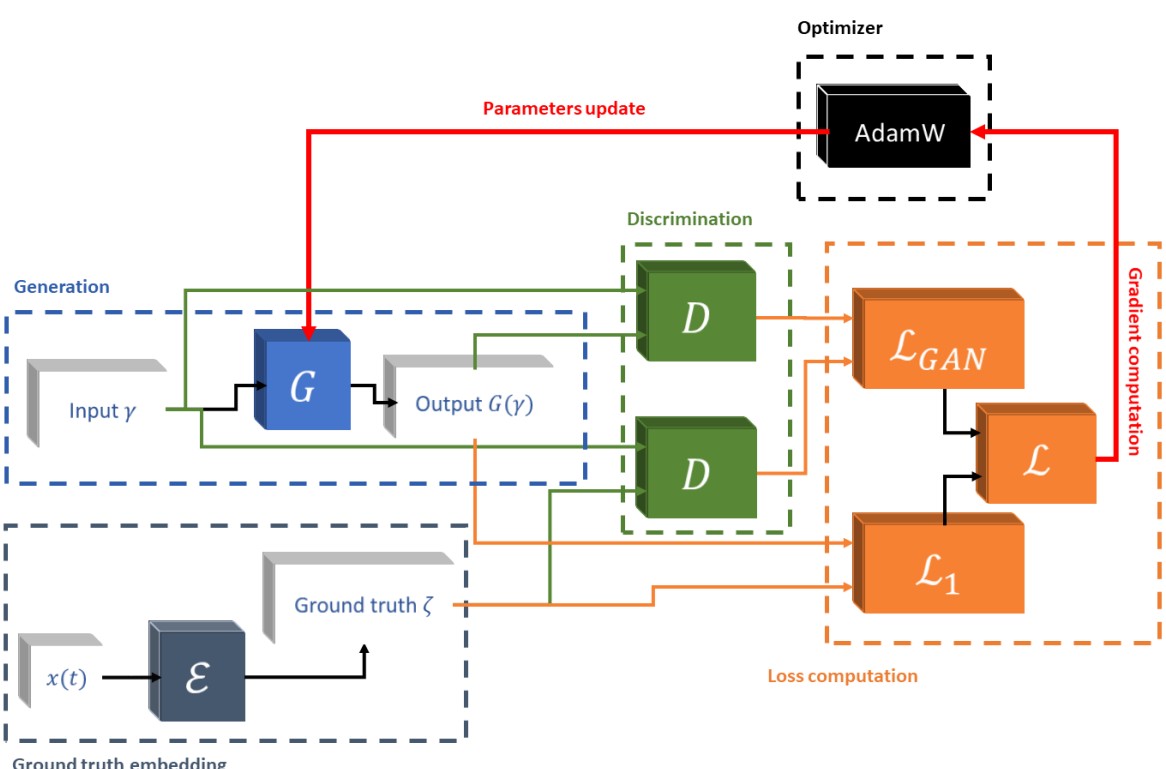

**Figure 3.** Training setup for the proposed network. In the case the GAN loss is not used, the discriminator and the $\mathcal{L}_{GAN}$ terms are ignored.





coordinate pair $x = (x_1, x_2)$ is embedded in an image $\{I\}_{i,j}$, $i, j = 1, \ldots, 24$ through a multivariate gaussian function centered in $x$. Let $x_{ij} \in [0,1]$ the normalized position of the $ij$-th pixel, then

$$I_{i,j} = \mathcal{E}(x)[i,j] \tag{5}$$
$$= \exp\left[-|x_{ij} - x|^2/\sigma^2\right]. \tag{6}$$

Setting $\sigma^2 = 1/2$ yields good results, albeit other options could be explored in the future.

The de-embedder $\mathcal{E}^{-1}$ is a deep convolutional neural network composed of 5 2D convolutional layers, each followed by a LeakyReLu non linearity, a flatten layer and a final 1D convolution feeded into a sigmoid activation function. The kernels are $3 \times 3$ for the first three convolutional layers, then $2 \times 2$ and $1 \times 1$, without padding. The kernel of the final 1D convolution is $1 \times 1$. The stride of the first two 2D convolutions is 2 (the strides of the rest being 1) and the number of filters of the 2D convolutions evolves as following: $16 \to 32 \to 64 \to 64 \to 64$. The 1D convolution has stride 1 and 2 filters, each outputting one of the two coordinates.

The loss function for the de-embedder training is

$$\mathcal{L}_{\text{EMB}} = \mathbb{E}_{x,z}\left[\left\|\mathcal{E}(x) - \mathcal{E}(\mathcal{E}^{-1}(\mathcal{E}(x) + z))\right\|_2^2\right] + \mathbb{E}_{x,z}\left[\left\|x - \mathcal{E}^{-1}(\mathcal{E}(x) + z)\right\|_2\right], \tag{7}$$

where $x \sim \text{Unif}(0,1) \times \text{Unif}(0,1)$. To increase the robustness of the de-embedder, a random noise $z$ is added with probability $p = 1/2$. Hence $z$ is either 0 or sampled from a normal random variable with $\mu = 0$ and $\sigma = 10^{-3}$, with probability $1/2$:

$$z \sim \begin{cases} \mathcal{N}(0, 10^{-4}) & \text{with prob. } 1/2 \\ \delta_0 & \text{with prob. } 1/2 \end{cases} \tag{8}$$

The loss function is designed to ensure that $\mathcal{E}^{-1}$ is both a left and a right inverse of $\mathcal{E}$. Furthermore, the second term directly evaluate the error in euclidean norm.

The de-embedder has been trained in a Monte-Carlo framework using the Adam optimizer, sampling random positions from $[0,1]^2$. Training parameters are: learning rate $\eta = 2 \cdot 10^{-2}$, $\beta_1 = 0.999$, $\beta_2 = 0.9999$, and AMSgrad option active. The training has been carried out in batches of 2000 samples for 50000 iterations, achieving an average error in euclidean norm of less than $2 \cdot 10^{-4}$.

Since the exact form of the embedding is known, non-machine-learning techniques have been considered, i.e. implementing the minimization of a suitable loss function using gradient descent. The results were satisfying in absence of noise, but even small jitters degraded the precision by several orders of magnitude. Hence we chosed the trained neural de-embedder as it resulted more robust with respect to noise and small variations.

Furthermore, we considered fine-tuning the neural de-embedder during the training of the Lagrangian prediction network, but the early results were disappointing: during the first stages of the training the de-embedder performances worsened notice-



ably, probably because it learned from non-gaussian images, thus decreasing precision. Furthermore, such fine-tuning might introduce bias, hence we decided not to pursue this strategy any further.

## 1.3 Methods

For the evaluation of the results we use three widely accepted methods for Lagrangian trajectories accuracy evaluation: root mean square error, Liu index Liu and Weisberg (2011) and Finite Size Lyapunov exponents Lacorata et al. (2019). Both Liu Index and FSLE evaluation techniques are specifically designed for Lagrangian simulations.

We give a brief description of the three techniques, referring to the original papers for a more detailed discussions.

Root mean square error is computed as

$$\mathrm{RMSE} = \left( \frac{1}{K} \frac{1}{T} \sum_{k=1}^{K} \sum_{t=1}^{T} d_k(t)^2 \right)^{1/2}, \tag{9}$$

where $d_k(t)$ is the distance between the $k$-th trajectory and its inferred counterpart at time $t$. The distance is computed using haversine distance.

For the $k$-th real-simulated trajectory pair Liu index is defined as

$$\mathrm{Liu}_k = \frac{\sum_{t=1}^{T} d_k(t)}{\sum_{t=1}^{T} l_k(t)}, \tag{10}$$

where $d_k(t)$ is defined as before and $l_k(t)$ is the length of the $k$-th (real) trajectory at time $t$. The single Liu indexes are then averaged to evaluate the whole inference. For both RMSE and Liu index, lower values indicate better simulation performance, higher values indicate worse simulation performance. Observe that, contrary to RMSE, Liu index is weighted by the distance traveled by the original particle, thus in some sense factoring out the size of the simulations, and allowing for better comparison between different simulations.

FSLE spectra were initially defined for Lagrangian study and characterization of dynamical regimes through couple statistics Boffetta et al. (2000). A certain number of pairs of initially close particles are left evolving under a dynamic, and their trajectories are then analyzed. In contrast to other methods, FSLE spectrum looks at separation properties along spatial scales, rather than along time scales. Essentially, FSLE measures the mean growth rate of a perturbation as function of its size $\delta$. More in detail, consider a starting scale $\delta_0$ and an amplification factor $\rho$, hence defining a set of $N$ scales $\delta_n = \rho^n \delta_0$, $n = 0, \ldots, N$. Then the FSLE spectrum is defined as

$$\lambda(\delta_n) = \frac{1}{\langle \tau_k(\delta_n) \rangle} \log(\rho) \tag{11}$$

,

where $\tau(\delta_n)$ is the time the $k$-th trajectory couple takes to separate from a distance of $\delta_n$ to a distance of $\delta_{n+1} = \rho \delta_n$ and $\langle \cdot \rangle$ denotes the average across all couples.





FSLEs allow for the characterization and quantification of the different dynamical regimes at different scales from a Lagrangian point of view. Tipically four major regimes are observed:

$$\lambda(\delta) \sim \begin{cases} \lambda & \text{Chaotic advection} \\ \delta^{-1} & \text{Ballistic dispersion} \\ \delta^{-2/3} & \text{Richardson diffusion} \\ \delta^{-2} & \text{Taylor diffusion} \end{cases} \tag{12}$$

.

The different regimes are related to the correlation between the underlying velocity field explored by the particles. At small scales, exponential separation typically occurs, resulting in chaotic advection. When the velocity difference is constant, i.e. two particles moving in the same coherent flow, the corresponding regime is ballistic dispersion. As the particles move further away from each other they typically enter in the Richardson turbulent diffusion regime, and when their velocities are completely uncorrelated the regime is standard Taylor diffusion, in which the separation between the two particles is statistically equivalent 235  to a Brownian motion.

    In Lacorata et al. (2019) FSLEs have been proposed as a tool to evaluate Lagrangian simulations and it has been used to compare different oceanic models in the Mediterrenean Sea. Two types of measures have been proposed: FSLE-I and FSLE-II. FSLE-I is the standard FSLE spectrum, computed separately for the real trajectory couples and simulated trajectory couples. Comparing the two FSLE spectra is informative in how much the simulated system is capable of reproducing the dynamicss of 240  the real system. FSLE-II on the other hand considers couples formed by real trajectory and simulated trajectory, starting from the same initial conditon. In this case the FSLE at a given scale $\delta$ indicates the frequency at which the simulation deviates from the real trajectory at that scale, i.e. the mean error growth rate at that scale.

    FSLE-II is expected to be, until a certain scale, in the regime of ballistic dispersion, meaning that the error growths across the scales as $\varepsilon(\delta) \sim \delta^{-\alpha}$ for $\delta \leq \delta^*$. Eventually, as the scale is large enough, simulation and real trajectory distance themselves 245  and the FSLE-II transitions into a turbulent or dissipative regime,

    In addition, in Lacorata et al. (2019, 2014), the mean error growth velocity $\gamma$, is defined as a function of the scale separation:

$$\gamma(\delta) = \delta\lambda(\delta)\frac{\rho - 1}{\log(\rho)}. \tag{13}$$

.

    It is straightforward that $\gamma(\delta)$ is the mean velocity at which simulated and real particles separate at scale $\delta$.

In Lacorata et al. (2019) FSLE-II and $\gamma(\delta)$ spectra have been used to diagnose in depth the effects of the application of submesoscale correction to different GCMs. For the sake of straightforward comparison of different simulations, we average $\gamma(\delta)$ across scales. Since the FSLE are computed on an exponential spatial scale, the average is weighted opportunely. We refer to this integrated measure as average separation velocity (ASV),



$$\text{ASV} = \frac{1}{\delta_{\bar{N}} - \delta_0} \int\limits_{\delta_0}^{\delta_{\bar{N}}} \gamma(\delta) d\delta \approx \frac{1}{\delta_{\bar{N}} - \delta_0} \sum_{k=0}^{\bar{N}} (\delta_{n+1} - \delta_n) \gamma(\delta_n) \tag{14}$$

where $\bar{N}$ is the last scale at which the FSLE spectrum is defined. Observe that both Liu and ASV measure the accuracy of
the simulation relative to the simulation scale, contrary to RMSE, which tends to increase as the simulation scale grows larger.

## 1.4  Results

The proposed network has been trained on synthetic trajectories generated from model-output surface velocities in the Mediter-
renean Sea for the year 2017, with resolution of 1/24 degrees. The duration of the Lagrangian simulations is 12 days, sampled
at 6 hours intervals, hence resulting in 48 time steps. The dataset is composed of trajectories simulated in two different time-
frames: 1-12 January and 3-15 September: from each simulation 40000 trajectories are randomly sampled, 32000 being added
to the training dataset and the rest added to the validation dataset, for a total of 64000 training examples.

For each time, the Eulerian fields have been divided in $24 \times 24$ tiles, with a $50\%$ overlap. The trajectories considered are
only those entirely contained inside a $24 \times 24$ tile. Note that this selection is done before sampling for the dataset.

The training has been performed for 1000 epochs in minibatches of 64 examples on an nVidia Tesla T4 gpu.

Two separate sets of training have been performed. In the first one, 8 time steps are considered for training, in the second
one 16 time steps have been considered instead. Due to hardware constraints, in the second case the size of the dataset has
been reduced to 20000 examples from each simulation, split as described before. For each configuration, training with both the
GAN and the non-GAN loss has been performed, resulting in 4 different trained networks. We identify these networks via the
training loss and the time steps used during training. Hence the networks are `gan 8`, `gan 16`, `nogan 8`, `nogan 16`.

During inference longer prediction times are obtained by iterating the application of the network: once the first prediction
segment is generated, the last position is taken as a new starting position for a new generation and so on. For each segment the
appropriate Eulerian fields tiles are selected according to the relative initial conditions. In this case the complete inclusion of
the generated trajectory in the tile is not guaranteed. To minimize the chances of exiting the tile, for each segment the Eulerian
fields tiles are chosen such that the trajectory initial conditions are in the center of the tile, i.e. if the lat-lon coordinates of tile
are indexed as $(x_{ij}, y_{ij})$, $i, j = 0, \ldots, 23$, then the initial conditions $(x(0), y(0))$ are contained in the square defined by indexes
$ij$ given by $[11, 11], [11, 12], [12, 11], [12, 12]$. The gaussian embedding is still needed to accurately encode the initial position.

Note that iterating the prediction as described before allows neural network approaches to Lagrangian simulation to be easily
scalable, avoiding the necessity of training on larger time frames in order to obtain long-term inferences.

Inference has been performed for three different time lengths: 8, 16 and 48 time steps, the latter being the entire length of
the trajectories in the dataset. Each inference set is defined by the network training method (`gan`/`nogan`), by the time length
of the training set (8/16) and by the inference time steps (8/16/48).

Two different periods have been considered for performance evaluation. The first one is the same period as one of the periods
considered in the training set, i.e. starting from the 1st to the 13th of January. The trajectories used for training are discarded





| 1st - 13th January | | | | | |
| Network | | | Metrics | | |
| Type | Tr. | Inf. | ASV [km/day] | Liu index [km] | RMSE [km] |
|---|---|---|---|---|---|
| gan | 8 | 8 | 10.44 | $0.392 \pm 0.068$ | $6.09 \pm 0.46$ |
| gan | 16 | 8 | 10.62 | $0.400 \pm 0.075$ | $\mathbf{6.02 \pm 0.46}$ |
| nogan | 8 | 8 | $\mathbf{8.65}$ | $\mathbf{0.383 \pm 0.072}$ | $5.97 \pm 0.48$ |
| nogan | 16 | 8 | 10.52 | $0.401 \pm 0.063$ | $6.04 \pm 0.46$ |
| gan | 8 | 16 | 3.34 | $0.204 \pm 0.031$ | $11.24 \pm 0.52$ |
| gan | 16 | 16 | $\mathbf{3.32}$ | $0.219 \pm 0.036$ | $12.21 \pm 0.51$ |
| nogan | 8 | 16 | 3.36 | $\mathbf{0.196 \pm 0.032}$ | $\mathbf{11.01 \pm 0.56}$ |
| nogan | 16 | 16 | 3.39 | $0.211 \pm 0.032$ | $12.11 \pm 0.53$ |
| gan | 8 | 48 | 0.89 | $0.096 \pm 0.015$ | $30.51 \pm 1.03$ |
| gan | 16 | 48 | $\mathbf{0.85}$ | $0.089 \pm 0.011$ | $\mathbf{28.54 \pm 0.93}$ |
| nogan | 8 | 48 | 0.92 | $0.094 \pm 0.016$ | $29.78 \pm 1.02$ |
| nogan | 16 | 48 | 0.88 | $\mathbf{0.086 \pm 0.012}$ | $28.61 \pm 0.90$ |

**Table 1.** Metrics for the time period 1st - 13th January. The different experiments are denoted by loss type, training time steps and inference time steps. Liu indexes and RMSE values have been computed through bootstrap technique, the values are given as average plus or minus two times the standard deviation. For each metric and each of the three different inference time steps, the best corresponding score is marked in bold.

and only never-seen-before trajectories are used. The second period is from the 21st of May to the 3rd of June. The number of trajectories used for the first period is $N_1 = 33400$, the number of trajectories used for the second period is $N_2 = 38400$. The results for the chosen metrics are reported in tables 1 (the first period) and 2 (the second period).

Both RMSE and Liu indexes and related confidence intervals have been computed using bootstrapping, the latter being estimated as two times the standard deviations.

Fsle have been computed using initial scale $\delta_0 = 5km$, with factor $\rho = \sqrt{2}$. To increase robustness in the computation of the ASV, in each of the experiments, the last scale in the FSLE spectrum was excluded, due to the low number of particle couples at those scales.

As can be seen in tables 1 and 2, the proposed network is capable of generating trajectories within acceptable error, even when the network is iterated for longer inference times. Observe that, according to tables 1 and 2, there is no substantial difference from gan training setup and non-gan training setup, except for the 8 steps training/8 steps inference models, where





21st May - 3rd June

| Network | | | Metrics | | |
|---|---|---|---|---|---|
| Type | Tr. | Inf. | ASV [km/day] | Liu index [km] | RMSE [km] |
| gan | 8 | 8 | 10.63 | $0.252 \pm 0.046$ | $5.89 \pm 0.39$ |
| gan | 16 | 8 | **8.91** | $0.261 \pm 0.059$ | $5.81 \pm 0.42$ |
| nogan | 8 | 8 | 8.93 | $\mathbf{0.241 \pm 0.057}$ | $\mathbf{5.79 \pm 0.45}$ |
| nogan | 16 | 8 | 9.38 | $0.267 \pm 0.069$ | $5.81 \pm 0.39$ |
| gan | 8 | 16 | 3.27 | $0.158 \pm 0.019$ | $11.19 \pm 0.48$ |
| gan | 16 | 16 | **3.20** | $0.164 \pm 0.027$ | $12.03 \pm 0.50$ |
| nogan | 8 | 16 | 3.22 | $\mathbf{0.151 \pm 0.019}$ | $\mathbf{10.84 \pm 0.49}$ |
| nogan | 16 | 16 | 3.39 | $0.162 \pm 0.023$ | $12.02 \pm 0.48$ |
| gan | 8 | 48 | 0.92 | $0.087 \pm 0.015$ | $31.22 \pm 0.89$ |
| gan | 16 | 48 | **0.84** | $\mathbf{0.078 \pm 0.011}$ | $\mathbf{28.83 \pm 0.89}$ |
| nogan | 8 | 48 | 0.91 | $0.082 \pm 0.012$ | $29.48 \pm 0.88$ |
| nogan | 16 | 48 | 0.85 | $0.081 \pm 0.015$ | $29.99 \pm 0.80$ |

**Table 2.** Metrics for the time period 21st May - 3rd June. The different experiments are denoted by loss type, training time steps and inference time steps. Liu indexes and RMSE values have been computed through bootstrap technique, the values are given as average plus or minus two times the standard deviation. For each metric and each of the three different inference time steps, the best corresponding score is marked in bold.

the non-gan is performing better than gan, in both the considered time periods. On the other hand, in the first time period, gan-trained models perform better in the ASV metric than non-gan when iterated, suggesting that gan training might be able to better generalize.

FSLE analysis provides better insight on the preformances of the different networks. In figures **??** and **??** the FSLE spectra
for the two different time steps training setups are shown (only 48 time-steps inference are shown). The FSLE-II's decay linearly ($\lambda(\delta) \sim \delta^{-1}$), in line with previous studies about simulations accuracy with FSLE (Lacorata et al. (2019, 2014)), and never reach turbulent or diffusive regime, respectively characterized by $\lambda(\delta) \sim \delta^{-3/2}$ and $\lambda(\delta) \sim \delta^{-2}$. This is mainly due to the inference time, which is short with respect to the time-scales where those regimes are present.

From the comparison of FSLE-I, it is clear that the gan-trained models better reproduce the dispersion characteristics. Fig.
**??** reports the FSLEs for the 8 time steps training time, for both time periods. Although less energetic at small scales than the original trajectories, the knees of the spectra of the gan-trained inferred trajectories are located at the same scale as the knee of





the FSLEs of the original trajectories. As the knee is where the separation between exponential and linear separation regimes occurs, hence the inferred trajectories preserve those dynamical characteristics. On the other hand, non-gan-trained models struggle at the task, as can be observed in fig. **??** (b) and **??** (c). The same considerations are valid when the training time is increased to 16 time steps, as shown in fig. **??**. In this case the non-gan models perform even worst than in the 8 time steps training setup, suggesting that training on longer times is not necessarily overly beneficial. Observe that the behavior is very similar across both time periods, indicating that the network has indeed correctly learned the relation between initial position, Eulerian fields and Lagrangian trajectory.

## 2 Conclusions

In this paper we presented a novel artificial neural network for Lagrangian trajectory prediction, based on the combination of convolutional LSTM and U-net. Inspired by existing computer vision approaches, the proposed network has been trained using both a weighted $\ell^1$ loss and an additional adversarial loss, in conditional generative adversarial network (cGan) fashion. The network has been trained and tested on simulated trajectories in the mediterrenean sea, and different training and inference times have been considered, iterating the network when training time is shorter than inference time. The inference experiments have been evaluated using Lagrangian metrics. As integrated metrics are not sufficient to show differences between the several proposed implementations, Finite Size Lyapunov Exponents (FSLEs) are analyzed. As a result, gan-trained models appear to be better capable of reproducing the dynamical characteristics of the underlying physics. Further research includes fine-tuning the networks on real trajectories, as well as train the network in different basins.

*Code and data availability.* The Eulerian tracers data used for training and trajectory generation are available on Zenodo Ciani (2024). The code for training and testing the neural network has been written in Python within the TensorFlow package framework Abadi et al. (2016), and is available on Zenodo, along with the trajectory dataset used Della Cioppa (2025).

*Author contributions.* LDC and BBN contributed equally to conceptualization, writing of the manuscript, reviewing and editing. LDC designed and coded the network, created the dataset and performed the trainings and the numerical evaluations of results. BBN provided funding for this research.

*Competing interests.* The authors declare no competing interests.

*Acknowledgements.* This work has been financially supported by the Italian national project ITINERIS (IR0000032 - ITINERIS, Italian Integrated Environmental Research Infrastructures System, D.D. n. 130/2022 - CUP B53C22002150006, Funded by EU - Next Generation





EU PNRR - Mission 4 "Education and Research" - Component 2: "From research to business" - Investment 3.1: "Fund for the realisation of an integrated system of research and innovation infrastructures")



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
