# Peer review of "Predicting oceanic Lagrangian trajectories with hybrid space-time CNN architecture"

_EGUsphere, 2025_

## Community Comment (CC1)

The paper introduces a modification of a CNN architecture to predict lagragian trajectories in the ocean. The work uses synthetic data simulated with a numerical model. Most of the effort of the work is focused on the presentation of the CNN architecture. The work concludes that the developed architecture correctly reproduces the underlying dynamics of the model.

The main contribution from the work lies on the development of the CNN architecture. Not much attention is paid to the underlying dynamics or to justify why the proposed method may be desirable over classical approaches. The paper's focus is more about deep learning methods than about geosciences modeling, which suggests that GMD may not be the most appropriate venue for this work.

*We acknowledge that the reviewer's concern originates from the lack of clarity on the description of the objectives and presentation of the results of our work in the present version of the manuscript. These can be substantially improved by revising the text. As it will appear evident from the detailed replies in the following, our work is specifically focused on improving Lagrangian modeling from partial knowledge of surface dynamics. It is not limited to ML development per se and has strong potential for future advancements in geoscientific applications. As such, we do not agree on the general suggestion to submit the paper to a ML-specific journal and we are confident that the reviewer will also agree on that point.*

On top of this misalignment, I have two main concerns, the first one with the rationale of the paper. In the paper's introduction, it is discussed that models are not able to capture all the relevant dynamics and that their predictions are then off with observations. However, then, the paper proceeds to derive lagragian trajectories from model's data. What would then be the gain? Why developing a deep learning method if the main information required to derive the trajectories is already computed by the model? This way to proceed (using DL methods to reproduce model results) is what is normally called an emulator or metamodel and tends to make sense when computational cost is a concern. I do not see the value of such approach in any other case, unless a compelling justification is provided.

*We thank the reviewer for pointing this out. In fact, we did not realize that the objectives of the work were not presented clearly in the present version of the manuscript. We agree that the rationale of our work does require a much clearer explanation, which we would be able to provide easily through a revision of the text.*

*Indeed, in the present Introduction we did not state clearly enough that our model aims to reconstruct the "true" trajectories (those that result from the full velocity field), but using as predictors only sequences of approximated (geostrophic) velocity fields and tracer distribution evolution.*

*We have basically set up an observing system simulation experiment where the full model is used to simulate the target trajectories and only partial information based on sea surface height and sea surface temperature is then used by the CNN to predict the trajectories. The experiment is designed specifically as a feasibility test, that we plan to extend in the future*

*directly to observations provided by satellite sensors (radar altimeters, SWOT, thermal infrared radiometers, optical instruments, etc.).*
*Lines 58-61 conveyed wrong information, not explaining clearly that: 1) we aim to reconstruct "true" trajectories (built from full dynamics) starting from partial surface information; 2) this choice is driven by the idea to successively test/fine-tune the model to real observation-based ocean surface measurements provided by satellites.*

*We are confident that this can be straightforwardly clarified in a modified text.*

The second concern refers to the methodology. Although I will provide more details below, I do miss many details about the input data used -which are the heart of any machine learning method- and I do miss a more robust methodology to compare the new method with classical approaches, to understand the value of the proposed approach. Therefore, my recommendation is to reject the paper. Below, I develop some of these points with more detail.

*Please find our reply to each point below.*

1. In my opinion, the paper is much more focused on the deep learning method developed than on the ocean dynamics that drive the trajectories mentioned in the title. This is not a problem in itself, but I wonder if a journal about geosciences modeling is the right outlet for the work. Almost all the methodological detail provided is related to the CNN used, and therefore, I believe that an artificial intelligence journal would be a much better outlet, where AI scientists will judge the method with much more rigor and knowledge that geoscientists may do.

*Based on the declared "Aims and scope" of GMD, our understanding is that it accepts contributions based on any kind of mathematical model that is adapted/developed to describe a specific geoscientific problem. Statistical models and neural networks are clearly possible candidates, and our convolutional neural network architecture has been designed to solve a specific geoscientific problem, namely the prediction of Lagrangian trajectories from partial knowledge of oceanic surface dynamics.*

*Moreover, we have also tested new methods to assess the model performance, based on FSLE of II type, which also clearly stands as a relevant topic for GMD.*

*As such, we cannot agree with the reviewer's suggestion. On the other hand, we are confident that a much clearer presentation of the objectives of the work and related results would possibly change the reviewer's opinion.*

2. The title and the introduction of the paper convey the idea that the model is going to be fed with real data, making the DL method a way to circumvent the need for modeling. Although I tend to dislike this kind of approach, I have seen methods in the geosciences (specially in atmospheric sciences) that outperform classical methods clearly. So, if the paper showed a DL method that was more able to forecast trajectories than classical

methods, I would have seen value in it. However, the work focuses on emulating the results of a model, which is less interesting, since the model results will always be required to feed the model. This emulation tends to make sense when computational cost is a concern, but it does not seem to be the case in this specific work.

It is also important to note here that model emulation is one of the tasks where machine learning shines, so I would not be surprised at all that it does also in this case. However, if this is the application that the authors have in mind, I would say that the most important analysis is about the proposed DL method, and thus, an AI journal would be the better place to get a robust review of the contribution.

*As explained above, our model is not an emulator of an advection scheme, and its advantage relies in its capability to improve Lagrangian diagnostics starting from approximated velocity fields and tracer evolution. This would be clarified in a revised version of the paper. Again, we cannot agree on the suggestion to submit to a generalistic AI journal, as the proposed model is relevant for geoscientific applications.*

3. The paper requires methodological improvements. Any new method should be compared with an old one for the reader to understand the advantages of the newly proposed method. In this case, the new method is presented and many metrics computed, but without a comparison, it is difficult to evaluate the real contribution of the newly proposed model. When working with model emulators, the comparison may be done using the DL method to compute trajectories generated with the model, but that the DL method has never seen before. In the specific case of this paper, I am not really sure how the comparison could be made, and that is why I mentioned above that I see a problem with the rationale of the work.

*Concerning the comparison with other models, we agree with the reviewer. In practice, however, no previous model has been proposed in the literature to retrieve Lagrangian trajectories from the partial data we are considering here. As such, we have rather compared different architectures showing the gain in performance provided by the selected final configuration.*

4. Much more detail is needed about the input data used to calibrate the method. Since ML model performance heavily relies on the data used, the paper should present those data in much more detail that it does. If possible, it would also be appreciated if a figure presenting the input data was created, so that the reader can have a better picture of the data used in training. The current description is not clear enough.

*The reviewer is right. These details would be easily included in a revised manuscript.*

5. Several figures seem to be missing in the manuscript, making it impossible to properly judge the results and conclusions of the paper. My recommendation is mostly based on the rationale, the main aim and the methodology, but I wanted to highlight this point anyway.

*The reviewer is totally right. We did not realize that the pdf version uploaded at the submission did not include all figures as it should, and we do apologize for that. Of course we can easily solve the issue and are ready to respond to further comments once the figures supporting our conclusions are properly shown.*

---

## Author Comment (AC1)

The authors have developed a neural network approach to predict Lagrangian trajectories. As the authors state, this is equivalent to solving an ordinary differential equation for a given integration time. The prediction of Lagrangian trajectories from known velocity fields is well-established with known inaccuracies from both the numerical integration, wind effects, and poorly resolved sub-mesoscale features when using geostrophic approximations. Their neural network approach requires specifying beforehand the length time of interest for both the training and the prediction. Presumably a new model would need to be trained for longer integration times, or their model would be iterated. The model requires the same velocity fields that conventional Lagrangian integration requires, in addition to temperature fields.

Given the reduced universality of their approach when compared to a simple numerical integration, the requirement of additional data sources, and the black-box nature of such machine learning, I see no reason why this method should be published. They have proposed a method with reduced utility, limited transparency, and no clear explanation why this is necessary or beneficial when compared with standard approaches.

*We appreciate the reviewer's feedback and recognize that their concerns stem primarily from insufficient clarity in the current manuscript regarding our objectives and methodology. We will address this through comprehensive revisions.*
*Crucially, our work does not aim to emulate conventional Lagrangian integration using full velocity fields. Instead, we focus on the real-world challenge of estimating trajectories from partial observations available via remote sensing: specifically, surface geostrophic velocities and sea surface temperature (SST) evolution. Full velocity fields, required by conventional methods, are not directly observable.*
*Our neural network leverages the synergistic information within these observable surface fields (geostrophic velocity and advection-driven SST patterns) to improve trajectory predictions beyond what's possible by simply integrating geostrophic velocities alone. This addresses a key limitation in observation-based Lagrangian diagnostics where sub-mesoscale processes and wind effects are poorly resolved.*
*Therefore:*

- *The requirement for SST data is a core advantage, not a limitation, as it provides physically relevant information unavailable to conventional integration using only geostrophic velocities.*
- *The method offers increased utility for observational applications, enabling improved trajectory estimates where full velocities are unknown.*

*We acknowledge the challenge of model interpretability inherent in ML approaches. In the revision, we will enhance discussion on this aspect and the physical insights learned by the network. We are confident that clearly presenting these objectives and results will resolve the reviewer's concerns regarding the necessity, benefit, and novelty of our approach compared to standard methods applied under observational constraints.*
General Comments:

The introduction is inadequate when reviewing earlier Lagrangian machine learning approaches.

*We will revise the Introduction by carrying out a deeper search of the most recent and relevant literature and we are obviously open to receive any suggestion from the reviewer on specific papers to be cited.*

*All specific comments reported below would be easily addressed during the revision,*

Specific Comments:

L16: Qualify this statement as there is a nonzero slip velocity between a drifter and the surrounding fluid.

L25-27: This under sampling needs quantification or citation

L38: Examples of such models?

L38-40: what metrics? You need a citation here too.

L40-41: Due to the chaotic nature of ocean surface currents, no model is capable of simulating Lagrangian trajectories without error. Please clarify what you mean by this.

L41-42: Remove this sentence

L47-57: This is far from a sufficient review of recent attempts to improve models of lagrangian trajectories with deep learning or neural networks. Please perform a meaningful literature review so as to put your results in context

L115-116: You require more data to perform a task which is already mathematically well-defined (equation 1) and has well-defined numerical accuracy.

Technical Comments:

Fix your citation style

L23-25 Rephrase this sentence for clarity